# Multi-site bone marrow core biopsy improves diagnostic accuracy in dogs with hematologic disease

Kristi M. Smiley[1], Sara L. Connolly[1], Rose Raskin[2], Michael F. Rosser[1], Amy N. Schnelle[1], Nicolas Lopez-Villalobos[3], Arnon Gal[1]*

1 Department of Veterinary Clinical Medicine, University of Illinois Urbana-Champaign, Urbana, Illinois, United States of America, 2 Department of Comparative Pathobiology, Purdue University, West Lafayette, Illinois, United States of America, 3 School of Agriculture and Environment, Massey University, Private Bag 11 222, Palmerston North, New Zealand

* agal2@illinois.edu

## Abstract

### Background

Spatial heterogeneity within bone marrow significantly affects diagnostic accuracy in human and veterinary medicine, where single-site sampling may fail to detect focal disease processes. However, optimal bone marrow sampling strategies for canine hematologic disease diagnosis remain unclear, representing a critical knowledge gap in veterinary diagnostic pathology. We hypothesized that multi-site bone marrow core sampling would provide superior diagnostic accuracy compared to current single-site sampling standards in canine patients. The primary aim was to evaluate diagnostic capture probability of sampling one to four bone marrow sites in dogs with suspected hematologic disease.

### Methods and findings

Sixteen dogs with suspected hematologic disease underwent bone marrow trephine biopsies from four anatomical locations (bilateral proximal humerus and iliac crest) using the ARROW OnControl Powered Driver system. Two board-certified clinical pathologists independently evaluated 64 masked bone marrow samples. Diagnostic accuracy was assessed using truth set methodology, with statistical analysis including mixed-effects logistic regression and bootstrap confidence intervals. Multi-site core sampling significantly improved diagnostic capture probability. Moving from one to two sites increased diagnostic accuracy from 76.6% to 94.8% under the permissive rule (18.2% improvement, P<0.0001), and from 28.1% to 47.9% when both pathologists agreed (19.8% improvement). Significant site-specific differences were observed in myeloid-to-erythroid ratios and megakaryocyte counts. Overall, 13.3% of

**Data availability statement:** The data used for analysis are available as a supplementary file (SUPP TABLE 1).

**Funding:** The authors received funding for this work from the University of Illinois Urbana-Champaign Companion Animal Research Grant, Wayne and Josephine Spangler Fund. The funders had no role in study design, data collection and analysis, decision to publish, or preparation of the manuscript.

**Competing interests:** The authors declare that no competing interests exist.

**Abbreviations: BM**, Bone marrow; **Bx Site**, Biopsy site; **CBC**, Complete blood count; **CI**, Confidence interval; **GLIMMIX**, Generalized linear mixed model; **ICC**, Intraclass correlation coefficient; **LH**, Left humerus; **LI**, Left iliac crest; **MER**, Myeloid-to-erythroid ratio; **OR**, Odds ratio; **RI**, Right iliac crest; **RH**, Right humerus; **SE**, Standard error.

samples were nondiagnostic, with modest inter-pathologist agreement ($\kappa = 0.30$, 42% agreement).

## Conclusions

Multi-site bone marrow core sampling provides clinically meaningful improvements in diagnostic accuracy for canine hematologic diseases, with the greatest benefit achieved by adding a second sampling site to current single-site protocols.

## Introduction

Bone marrow evaluation represents a cornerstone diagnostic procedure in veterinary hematology, providing essential insights into hematopoietic disorders that cannot be adequately assessed through peripheral blood analysis alone. [1–3] In dogs, bone marrow examination is primarily indicated for investigating persistent cytopenias, abnormal cell morphologies, suspected neoplastic processes, and unexplained hematologic abnormalities.[1,4–11] The procedure involves both aspiration and core biopsy collection, with samples typically obtained from anatomically accessible sites, including the proximal humerus, iliac crest, proximal femur, and sternum. [12–14] Despite its clinical importance, current veterinary practice predominantly relies on single-site sampling protocols, which may inadequately capture the full diagnostic potential of bone marrow evaluation due to inherent spatial heterogeneity within hematopoietic tissues. [3,6,7,10,15]

Recent advances in human medicine have increasingly recognized the phenomenon of spatial heterogeneity within bone marrow, where cellular composition, disease processes, and pathological features can vary significantly across different anatomical locations within the same patient. [16–18] This heterogeneity has profound implications for diagnostic accuracy, as single-site sampling may fail to detect focal disease processes or provide representative samples of the overall marrow status. [19] In multiple myeloma, studies have demonstrated that malignant plasma cells exhibit spatially restricted distributions, with significant genomic and phenotypic differences observed between bone marrow sites and focal lesions. [20,21] Similarly, investigations using spatial transcriptomics have revealed complex microenvironmental gradients within bone marrow, with distinct cellular niches and signaling domains that support different hematopoietic populations. [22]

The limitations of single-site sampling have been highlighted in human hematologic oncology, where a limited number of studies demonstrated that multi-region sampling can improve the detection of focal or spatially heterogeneous disease compared to conventional single-site methods. [23] These findings support the broader understanding that hematopoietic disorders may exhibit regionally distributed lesions rather than uniform involvement, suggesting that multi-site sampling could enhance diagnostic accuracy in selected human conditions. [24,25] In veterinary hematology, however, single-site sampling protocols remain standard practice, [10] and the

potential diagnostic benefit of multi-site approaches for canine bone marrow evaluation has not been systematically investigated. [12,13,26]

The diagnostic accuracy of bone marrow evaluation is further complicated by technical factors including sample quality, cellular composition variability, and inter-observer interpretation differences. [1,12,13,27] Studies in both human and veterinary medicine have reported significant variability in sample adequacy, with non-diagnostic rates ranging from 8–15% depending on sampling technique and anatomical location. [12,13] Additionally, site-specific differences in cellular composition, including variations in myeloid-to-erythroid ratios, megakaryocyte numbers, and iron content, have been documented, suggesting that anatomical location significantly influences both sample characteristics and diagnostic interpretability. [13,28–30] Gal et al. demonstrated that the site of bone marrow acquisition significantly affects the myeloid-to-erythroid ratio in apparently healthy dogs, with consistent differences observed between anatomical locations. [28] Similarly, variations in bone marrow iron stores have been documented across sampling sites in dogs, with implications for the interpretation of iron-restricted erythropoiesis. [13,29]

Previous veterinary studies have provided limited evidence supporting multi-site sampling approaches. Abrams-Ogg et al. compared canine core bone marrow biopsies from multiple sites using different techniques and needles, demonstrating variability in sample quality and cellular composition between anatomical locations. [12] Defarges et al. compared sternal, iliac, and humeral bone marrow aspiration in Beagle dogs, revealing site-dependent differences in sample characteristics. [13] Furthermore, Aubry et al. evaluated bone marrow aspirates from multiple sites for staging of canine lymphoma and mast cell tumors, suggesting that multi-site sampling may improve diagnostic accuracy in neoplastic conditions. [26] However, these studies have been limited by small sample sizes and focus on specific disease conditions rather than a comprehensive evaluation of diagnostic accuracy across the spectrum of hematologic diseases encountered in clinical practice.

Given the critical role of bone marrow evaluation in canine hematologic disease diagnosis and the emerging evidence for spatial heterogeneity in hematopoietic tissues, there exists a significant knowledge gap regarding the optimal sampling strategy for veterinary bone marrow examination. The potential for improved diagnostic accuracy through multi-site sampling approaches, as demonstrated in selected conditions in human medicine, warrants systematic investigation in veterinary patients to establish evidence-based protocols that maximize diagnostic yield while maintaining procedural safety and feasibility. [12–14,26]

We hypothesized that multi-site bone marrow core sampling would provide superior diagnostic accuracy compared to current single-site sampling standards in canine hematology, and that anatomical location would significantly influence sample quality and diagnostic interpretability. The primary aim of this study was to evaluate the diagnostic capture probability of sampling one to four bone marrow core biopsy sites in dogs with suspected hematologic disease. Secondary objectives included assessing site-specific differences in sample quality, cellular composition, and inter-pathologist agreement, while determining the optimal number and combination of sampling sites to maximize diagnostic accuracy in clinical veterinary practice.

## Methods

This randomized, blinded clinical trial, carried out at the University of Illinois Veterinary Teaching Hospital, prospectively enrolled client-owned dogs needing bone marrow (BM) evaluation as part of their diagnostic assessment for hematologic illness between July 2020 and April 2023. The research received approval from the University of Illinois Institutional Animal Care and Use Committee (IACUC #19119), and the owners granted informed written consent at the point of study enrollment. Criteria for inclusion were a body weight of more than 7 kg and clinical indications for BM examination, such as unexplained cytopenias or suspected marrow-based infectious, neoplastic, or immune-mediated disorders. All owners gave written informed consent prior to participation, and all procedures were performed by a board-certified veterinary internist (AG).

The ARROW OnControl Powered Driver system (Teleflex, Morrisville, NC, USA) was used to streamline the BM collection and reduce the duration of the procedure. Each dog underwent BM evaluation from four anatomical locations, the right and left proximal sections of the humerus and the iliac crest on both sides, according to a randomized block schedule (see **Table 1**). The rationale for randomization was to eliminate potential sequence effects, such as changes in operator technique, fatigue, or minor procedural variability that could systematically bias sample quality or diagnostic yield at specific sites. At the time of marrow sampling, a complete blood count (CBC) including a differential was obtained to enhance histopathological interpretation.

Sampling of the iliac crest was performed at the widest cranial-dorsal section of the wing, with the needle inserted parallel to its long axis. For the humeral sites, the needle was placed perpendicular to the craniolateral bone surface, lateral and distal to the greater tubercle. Core bone biopsy samples were collected using an 11-gauge BM trephine biopsy needle (OnControl system). After sample collection, all skin incisions were sealed with tissue adhesive. The BM trephine biopsies were fixed in 10% buffered formalin for 24h, followed by 24h of decalcification in 5% aqueous solution of EDTA disodium salt dihydrate. The UIUC Veterinary Histology Laboratory was contracted to process the formalin-fixed decalcified BM trephine biopsies, embed them in paraffin wax, section 3-µm thick histological sections, and stain samples from each site with hematoxylin and eosin, periodic acid–Schiff and Giemsa (utilizing both stains allowed for better differentiation between erythroid, myeloid and lymphoid cells), impregnated silver stain for reticulin, and Prussian blue stain for iron. Immunohistochemical stains were not performed.

Each specimen received a randomized accession identifier to guarantee that dog identity and sampling site were concealed from the two experienced veterinary clinical pathologists raters. To maintain an unbiased assessment, clinical pathologists remained blinded to all samples throughout slide preparation and analysis.

Primary Outcome: site-specific clinical diagnoses were established based on BM histopathology in the context of CBC results. For data analysis, every site within a given dog was assigned a numeric code as per a preset scheme, and up to three diagnostic codes could be attributed to each site by the reviewing clinical pathologists (see **Table 2**, **S1 Table**).

**Table 1. Sequence of sampling (randomized complete block design).**

| Dog No. | 1st | 2nd | 3rd | 4th |
|---|---|---|---|---|
| 1 | LI | LH | RH | RI |
| 2 | LI | RH | LH | RI |
| 3 | LH | RH | LI | RI |
| 4 | LH | LI | RI | RH |
| 5 | RH | LI | RI | LH |
| 6 | LH | RI | LI | RH |
| 7 | LH | RI | RH | LI |
| 8 | LI | RI | LH | RH |
| 9 | LI | RH | RI | LH |
| 10 | LI | LH | RI | RH |
| 11 | RH | RI | LH | LI |
| 12 | LH | RH | RI | LI |
| 13 | RI | LH | LI | RH |
| 14 | RH | RI | LI | LH |
| 15 | LI | RI | RH | LH |
| 16 | LH | LI | RH | RI |

Abbreviations: LH, left humerus; RH, right humerus; LI, left iliac crest; RI, right iliac crest.

**Table 2. Diagnostic code list.**

| Code No. | Explanation |
|---|---|
| 1 | Hyperplasia of one or more than one lineage |
| 2 | Myelodysplastic syndrome |
| 3 | Leukemia/ round cell neoplasia |
| 4 | Hypoplasia of one or more than one lineage |
| 5 | BM inflammation/ infection |
| 6 | BM toxicity |
| 7 | BM fibrosis |
| 8 | Metastatic neoplasia |
| 9 | Non-diagnostic sample |
| 10 | Other |

Secondary Outcomes: secondary measures included percent BM cellularity, myeloid-to-erythroid ratio (MER; based on a 300-cell differential), blast cell percentage (out of 300 cells), average megakaryocyte count (based on 10 random 40 × microscopic fields), ordinal assessment of overall sample quality (scored as non-diagnostic [=1], poor [=2], adequate [=3], or excellent [=4]), 1–4 ordinal iron stores score (1 = absent; 2 = minimal [rare, small foci of faintly visible granules]; 3 = moderate [clearly visible multifocal deposits in macrophages or along trabeculae]; 4 = abundant [dense, coalescing granules widely distributed throughout the marrow], based on 10 random 40 × microscopic fields), and whether lymphocytosis and plasmacytosis exceeded 5% and 2%, respectively.

## Power sample size estimation

To ensure the study had adequate statistical power, two investigators (AG and NLV) conducted a power analysis using bootstrap simulations in R software (version 3.5.1). This analysis involved simulating 10 replicate experiments, with each experiment consisting of 1,000 resamples. In each simulation, a cohort of 15 dogs was created, and discordant BM results were generated using a Bernoulli process, assuming a 5% discordance rate based on previous findings in healthy dogs. For each simulated dog, sampling from four anatomical sites was modeled with two independent ratings per site, resulting in eight data points per animal. The simulated discordance rate from each resample was then tested against a clinically expected discordance rate of 10% using a chi-squared test with a significance level of α = 0.05. The results of this simulation indicated that a sample size of 15 dogs would provide approximately 91% power (95% CI: 90.2%–91.4%) to detect a significant difference between the observed and expected rates of discordance. Anticipating a non-diagnostic rate of around 33% based on prior work, a target enrollment of 20 dogs was initially set to account for potential sample attrition. The final study ultimately included 16 dogs that met all inclusion criteria.

## Statistical methods

Statistical analyses were conducted by two investigators (AG and NLV) using SAS version 9.4 (SAS Institute Inc., Cary, NC). Two board-certified clinical pathologists (RR and SC) independently rated masked bone marrow biopsy cores. Non-diagnostic reads (histopathology code = 9) were summarized and then excluded from primary analyses. To judge accuracy without an external gold standard, a dog-specific "truth set" was created using the histopathology code(s) most frequently observed across that dog's sites and both pathologists (δ = 0; ties retained) (**S2 Table**). For each dog–site pair, we evaluated three truth-set consistency rules: ANY (at that site, at least one pathologist's interpretation is in the truth set), BOTH_ any (at that site, both pathologists' interpretations are in the truth set; if the truth set contains ties, the two interpretations may differ), and BOTH_same (at that site, both interpretations are in the truth set and both pathologists selected the same truth-set code; this agreement requirement applies even when the truth set contains ties). A read was deemed correct

when its code belonged to the dog's truth set. To determine how many of the 4 sites were correct, the probability of capturing $\geq 1$ correct site was computed when sampling k = 1–4 sites ($m \in [0, 1, 2, 3, 4]$, $P(\geq 1) = 1 - P(0\ correct) = 1 - \frac{\binom{4-m}{k}}{\binom{4}{k}}$). These probabilities were averaged across dogs, and 95% CIs were obtained using 2,000 resamples under bootstrapping sampling (stratified by rule and k); paired differences between successive k values were bootstrapped similarly, and medians were tested against 0 with nonparametric location tests. To evaluate site effects on correctness, a mixed-effects logistic regression was fitted with biopsy site and pathologist as fixed effects and a random intercept for dog; Tukey adjustment was used for pairwise site comparisons, and model-based probabilities with 95% CIs are reported. Inter-pathologist agreement was assessed overall and by site using Cohen's kappa and percent agreement with exact (Clopper–Pearson) 95% CIs; category levels were restricted to those observed by both readers to ensure valid tables. Intra-observer agreement (within pathologist, across site pairs within dog) used pairwise κ and percent agreement with exact 95% CIs; Light's κ (mean of pairwise κ) summarized within-reader agreement.

## Results

Among 128 possible bone-marrow interpretations (16 dogs × 4 sites × 2 clinical pathologists), 18 (13.3%) were nondiagnostic and occurred in 7 dogs; most nondiagnostic reads were from the right ilium and right humerus, and pathologist 1 contributed two-thirds of these calls (**Table 3**).

Two core biopsies (RH and LI from two different dogs) could not be obtained (missing data). Overall code frequencies are summarized in **Table 3**. Code 3 (leukemia/round-cell neoplasia) was assigned in 16 of the BM interpretations in 6/16 dogs (37.5%). Among these, 3 dogs had multi-site involvement (2 dogs with concordant Code-3 calls by both pathologists across ≥2 sites; 1 dog with multi-site Code-3 calls from one pathologist only). The remaining 3 dogs had single-site Code-3 assignments (1 dog concordant across both pathologists; 2 dogs called by one pathologist only). The per-dog site count for Code 3 ranged from 1 to 3 of 4; no dog was Code 3 at all four sites. The chance of obtaining at least one truth-consistent site increased with the number of sites sampled (**Table 4**, **Fig 1**).

Moving from one to two sites produced the largest gain across all rules, with smaller increments thereafter, particularly under the permissive ANY rule (**Table 5**).

In a mixed-effects logistic model, neither site nor pathologist influenced per-read correctness; model-based site probabilities had wide, overlapping CIs, and all pairwise comparisons were non-significant (**Table 6**, **Fig 2**).

**Table 3. Diagnostic code distribution and nondiagnostic summary.**

| Category | Code | n/128 | % |
|---|---|---|---|
| BM hyperplasia | 1 | 38 | 28.1 |
| BM hypoplasia | 4 | 27 | 20.0 |
| BM fibrosis | 7 | 18 | 13.3 |
| Leukemia/ round cell neoplasia | 3 | 16 | 11.9 |
| BM inflammation | 5 | 4 | 3.0 |
| BM toxicity | 6 | 2 | 1.5 |
| Myelodysplastic syndrome | 2 | 1 | 0.7 |
| Nondiagnostic sample | 9 | 18 | 13.3 |
| Metastatic neoplasia | 8 | 0 | 0 |
| Missing data | — | 4 | 3.1 |

Nondiagnostic distribution. Dogs affected: 7/16; median within-dog nondiagnostic proportion 0% (range 0–50%); By site (of 18 nondiagnostic sites total): RI 7/18, RH 5/18, LI 5/18, LH 1/18; by pathologist: Path 1 = 12, Path 2 = 6.

**Table 4. Mean probability (across dogs) of obtaining ≥1 truth-consistent site when sampling *k* sites (without replacement) from four sites under three evaluation rules; values are means across dogs with 95% bootstrap CIs (2.5th–97.5th percentiles).**

| Rule | k = 1 | k = 2 | k = 3 | k = 4 |
|---|---|---|---|---|
| ANY§ | 76.6% (58.7–90.5) | 94.8% (81.5–100) | 98.4% (92.5–100) | 100% (100–100) |
| BOTH_any¶ | 37.5% (21.5–54.4) | 62.5% (37.0–82.3) | 78.1% (48.8–94.4) | 87.5% (55.3–100) |
| BOTH_same‡ | 28.1% (12.5–46.3) | 47.9% (23.1–72.8) | 62.5% (32.0–85.3) | 75.0% (39.3–97.4) |

§ANY: at a given site, at least one pathologist's interpretation is in the truth set.

¶BOTH_any: at a given site, both pathologists' interpretations are in the truth set; if the truth set contains ties, the two interpretations may differ.

‡BOTH_same: at a given site, both interpretations are in the truth set and the pathologists selected the same truth-set code; this agreement requirement applies even when the truth set contains ties.

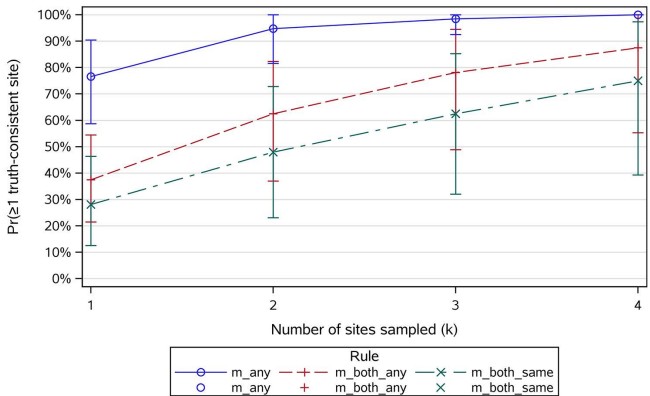

**Fig 1. Probability of capturing ≥1 truth-consistent site vs. number of sites sampled (k), by rule.** Lines show means across dogs; bands show 95% bootstrap CIs. m_any = ≥1 pathologist's read in the truth set; m_both_any = both in the truth set (not necessarily equal); m_both_same = both agree and the agreed code is in the truth set.

**Table 5. Paired, within-dog differences in p(k) for adjacent contrasts (bootstrap 95% CIs) and location tests.**

| Rule | Contrast | Mean Δp | 95% CI | t-test p | Sign p | Signed-rank p |
|---|---|---|---|---|---|---|
| ANY | 2–1 | +18.2% | 8.0–25.7 | <0.0001 | 0.0010 | 0.0010 |
| ANY | 3–2 | +3.6% | 0.0–11.0 | 0.0895 | 0.2500 | 0.2500 |
| ANY | 4–3 | +1.6% | 0.0–7.5 | 0.3332 | 1.0000 | 1.0000 |
| BOTH_any | 2–1 | +25.0% | 15.6–30.8 | <0.0001 | 0.0001 | <0.0001 |
| BOTH_any | 3–2 | +15.6% | 7.9–21.7 | <0.0001 | 0.0005 | <0.0001 |
| BOTH_any | 4–3 | +9.4% | 2.1–19.2 | 0.0090 | 0.0313 | 0.0313 |
| BOTH_same | 2–1 | +19.8% | 9.8–26.9 | <0.0001 | 0.0005 | <0.0001 |
| BOTH_same | 3–2 | +14.6% | 5.9–22.1 | 0.0002 | 0.0020 | 0.0020 |
| BOTH_same | 4–3 | +12.5% | 3.9–21.1 | 0.0015 | 0.0078 | 0.0078 |

p-values: t-test p = paired t-test (mean Δp). Signed-rank p = Wilcoxon signed-rank test (median Δp). Sign p = exact binomial sign test comparing counts of Δp > 0 vs Δp < 0 (ties excluded). Results were concordant; Wilcoxon is the prespecified primary test.

**Table 6. Per-read correctness by site (GLIMMIX least-squares means on the probability scale).**

| Site | Predicted probability | 95% CI |
|------|----------------------|--------|
| LH | 64.6% | 52.9–74.7 |
| LI | 56.2% | 38.9–72.1 |
| RH | 76.1% | 49.4–91.2 |
| RI | 60.1% | 40.2–77.1 |

Type III tests for site P = 0.599; Pathologist P = 0.781. Tukey-adjusted pairwise comparisons: all non-significant.

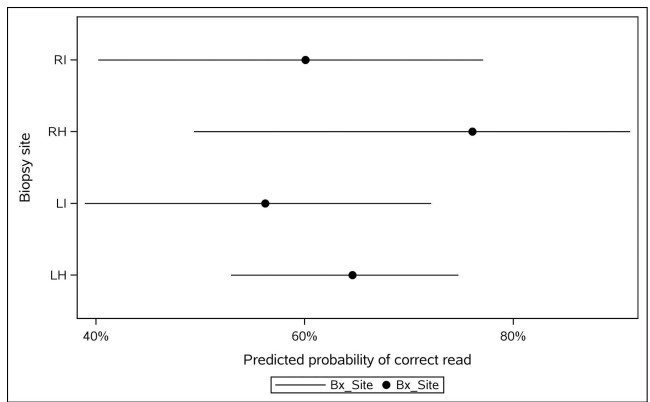

**Fig 2. Model-based site effects on per-reading correctness.** GLIMMIX least-squares means with 95% CIs for LH, LI, RH, RI on the probability scale. No site effect (p = 0.599) and no pathologist effect (p = 0.781). **Point**: LS-mean predicted probability; **Horizontal line**: 95% CI.

Inter-pathologist agreement was modest overall (κ = 0.30, 95% CI 0.11–0.49; 42% agreement, 95% CI 28.2–56.8%) and varied by site, being highest at RH and lowest at LH (**Table 7**).

Intra-observer agreement ranged from poor to substantial, depending on the site pair, with Pathologist 1 strongest at LH–RH and Pathologist 2 showing fair-to-moderate agreement across pairs; Light's κ indicated overall fair repeatability for both (**Table 8**).

When pooled across pathologists, the highest agreement remained for LH–RH and LH–RI, with the lowest for LI–RH and RH–RI (**Table 9**, **Fig 3**).

Secondary variables were compared across the four anatomical sampling sites (**Table 10**).

The MER, assessed by a single pathologist, showed significant site-dependent differences, with the highest ratios consistently observed at the right ilium. Least-squares mean logMER values were lower at the left humerus (LH) and right humerus (RH) compared to the right ilium (RI). Pairwise site comparisons confirmed that logMER was significantly

**Table 7. Inter-pathologist percent agreement by site (exact 95% CIs).**

| Site | Percent agree | 95% CI |
|------|---------------|--------|
| LH | 20.0% | 4.3–48.1 |
| LI | 27.3% | 6.0–61.0 |
| RH | 83.3% | 51.6–97.9 |
| RI | 41.7% | 15.2–72.3 |

Site-wise exact binomial CIs for the proportion of matching codes between pathologists.

**Table 8. Intra-observer agreement by site pair and pathologist.**

| Pathologist | Site Pair | Simple κ (95% CI) | % Agreement (95% CI) |
|---|---|---|---|
| 1 | LH–RH | 0.73 (0.41–1.00) | 81.8% (48.2–97.7) |
| 1 | LH–RI | 0.28 (−0.18–0.74) | 55.6% (21.2–86.3) |
| 1 | LI–RI | 0.25 (−0.40–0.90) | 62.5% (24.5–91.5) |
| 1 | LH–LI | 0.14 (−0.34–0.61) | 50.0% (15.7–84.3) |
| 1 | RH–RI | −0.02 (−0.45–0.41) | 33.3% (7.5–70.1) |
| 1 | LI–RH | −0.20 (−0.88–0.48) | 33.3% (4.3–77.7) |
| 2 | LH–RI | 0.47 (0.09–0.85) | 63.6% (30.8–89.1) |
| 2 | LH–RH | 0.39 (0.08–0.70) | 50.0% (21.1–78.9) |
| 2 | LH–LI | 0.38 (0.06–0.69) | 54.5% (23.4–83.3) |
| 2 | LI–RI | 0.34 (0.05–0.64) | 46.2% (19.2–74.9) |
| 2 | LI–RH | 0.27 (−0.09–0.63) | 45.5% (16.7–76.6) |
| 2 | RH–RI | 0.38 (0.03–0.72) | 50.0% (18.7–81.3) |

Light's κ: Path 1 = 0.28; Path 2 = 0.37. Confidence intervals that cross zero are not statistically significant (i.e., the observed agreement is not distinguishable from what would be expected by chance).

**Table 9. Pooled (across pathologists) intra-observer agreement by site pair.**

| Pair | Simple κ | 95% CI | % Agree | 95% CI |
|---|---|---|---|---|
| LH–RH | 0.54 | 0.29–0.79 | 65.2% | 42.7–83.6 |
| LH–RI | 0.47 | 0.18–0.75 | 60.0% | 36.1–80.9 |
| LH–LI | 0.33 | 0.04–0.63 | 52.6% | 28.9–75.6 |
| LI–RI | 0.36 | 0.07–0.65 | 52.4% | 29.8–74.3 |
| LI–RH | 0.17 | −0.14–0.48 | 41.2% | 18.4–67.1 |
| RH–RI | 0.22 | −0.08–0.52 | 42.1% | 20.3–66.5 |

Symmetry tests for pairwise tables were non-significant. Confidence intervals that cross zero are not statistically significant (i.e., the observed agreement is not distinguishable from what would be expected by chance).

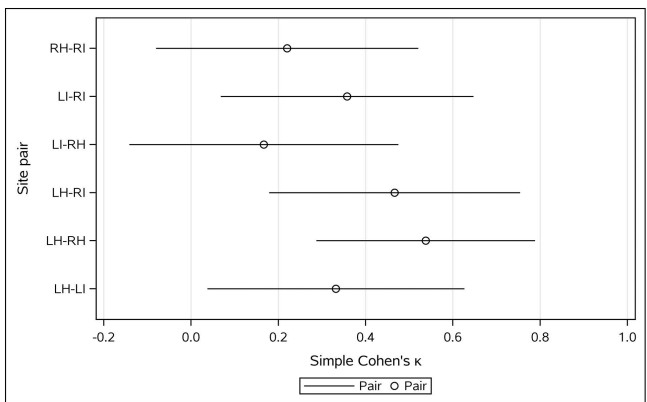

**Fig 3. Inter-pathologist agreement by site pair.** Simple κ with 95% CIs for each site pair pooled across pathologists, with N overlaid. Highlights that LI–RH shows the highest agreement. **Open circle**: κ estimate; **Horizontal line**: 95% CI.

**Table 10. Mixed-effects models: main effects and clustering by dog.**

| Outcome (dependent variable) | n | Biopsy site (Type III p) | Pathologist (Type III p) | Site×Path (Type III p) | ICC |
|---|---|---|---|---|---|
| logMER | 49 | 0.0102 | — | — | 0.54 |
| Log(Mean megas) | 100 | 0.0026 | 0.9477 | 0.6946 | 0.66 |
| Logit(%BM cells) | 102 | 0.5687 | <0.0001 | 0.6048 | 0.24 |
| Log(%blasts) | 106 | 0.0091 | <0.0001 | 0.0560 | 0.14 |

Type III p-values are from PROC MIXED

ICC = Var(Dog_ID)/ (Var(Dog_ID)+Residual); higher ICC means more between-dog clustering.

reduced at LH (p = 0.0309) and RH (p = 0.0092) when contrasted with RI, while differences among LH, LI, and RH were not significant. There was no lateral (left–right) asymmetry between the LI–RI (p = 0.1375) or LH–RH (p = 0.9135). Sampling site significantly affected the mean megakaryocyte numbers (p = 0.0026), with LI samples containing significantly fewer megakaryocytes than LH (p = 0.0084) and RH (p = 0.0034); RI did not differ significantly from the other sites. Blast percentage, but not percentage cellularity, differed significantly between sites, with humeri generally having a higher blast percentage (LH–RI, p = 0.0354; LH-LI, p = 0.0628; RH–RI, p = 0.0877; RH–LI, p = 0.1331). Notably, substantial inter-dog variability was observed for all marrow features. For MER, the calculated intraclass correlation coefficient (ICC) was 0.54, indicating that over half of the total variance in logMER was attributable to differences between dogs; the inter-dog coefficient of variation was 46%, reflecting considerable biological diversity among individuals.

The likelihood of lymphocytosis and plasmacytosis was evaluated across bone marrow sampling sites in dogs (**Table 11**).

For both outcomes, there were no significant site differences, but the probability for lymphocytosis differed significantly between the raters (p < 0.0001). Individual dog differences accounted for a substantial portion of variation for lymphocytosis (ICC = 0.58), indicating that more than half of the variability in lymphocytosis could be attributed to differences between dogs. For plasmacytosis, dog-level differences were less pronounced (ICC = 0.20), reflecting lower inter-dog variability for this parameter.

Lastly, the sample quality and iron stores ordinal scores were evaluated across the bone marrow sites (Table 12).

There were significant differences among sites for both outcomes, with sites having significant effects on sample quality (p = 0.002) and iron store scoring (p = 0.004). Pairwise contrasts from mixed-effects ordinal models showed that ilial samples (left and right ilium) were consistently of higher quality and had greater iron stores compared to humeral samples (left and right humerus). There was considerable variability between dogs for both sample quality and iron stores. Dog-level clustering was modest for sample quality (ICC = 0.02; dog-level variance = 0.06), but substantial for iron stores (ICC = 0.70; dog-level variance = 7.54), indicating strong between-dog differences in iron store scores and reflecting important

**Table 11. Mixed-effects logistic models: main effects and clustering by dog.**

| Outcome | Fixed effect | Num DF | Den DF | F value | p-value |
|---|---|---|---|---|---|
| Lymphocytosis (Yes/No) | Bx_Site | 3 | 83 | 1.66 | 0.1809 |
| | Path | 1 | 83 | 17.64 | <0.0001 |
| | Site×Path | 3 | 83 | 0.16 | 0.9213 |
| Plasmacytosis (Yes/No) | Bx_Site | 3 | 83 | 1.81 | 0.1522 |
| | Path | 1 | 83 | 0.02 | 0.8992 |
| | Site×Path | 3 | 83 | 1.15 | 0.3332 |

Type III tests of fixed effects from mixed-effects logistic regression (binomial, logit link) with a random intercept for dog (Dog_ID). Models were fit with PROC GLIMMIX (Laplace approximation; containment degrees of freedom). Outcomes are the presence/absence of lymphocytosis or plasmacytosis.

Abbreviations: **Bx_Site** = biopsy site; **Path** = pathologist; **Num DF** = numerator degrees of freedom for the test; **Den DF** = denominator degrees of freedom.

**Table 12. Pairwise site comparisons for ordinal scale outcome variables (cumulative logit models), with dog-level clustering (ICC).**

| Outcome variable | Contrast | Log OR | SE | p-value | OR | 95% CI for OR |
|---|---|---|---|---|---|---|
| Sample quality | LH vs LI | −1.35 | 0.52 | 0.0107 | 0.26 | 0.09–0.73 |
| | LH vs RH | 0.62 | 0.54 | 0.2492 | 1.86 | 0.65–5.37 |
| | LH vs RI | −1.21 | 0.51 | 0.0191 | 0.30 | 0.11–0.82 |
| | LI vs RH | 1.97 | 0.58 | 0.0011 | 7.15 | 2.26–22.66 |
| | LI vs RI | 0.14 | 0.52 | 0.7879 | 1.15 | 0.41–3.24 |
| | RH vs RI | −1.83 | 0.57 | 0.0018 | 0.16 | 0.05–0.50 |
| Iron Stores | LH vs LI | −2.37 | 0.68 | 0.0009 | 0.09 | 0.02–0.37 |
| | LH vs RH | −0.40 | 0.61 | 0.5106 | 0.67 | 0.20–2.24 |
| | LH vs RI | −1.67 | 0.62 | 0.0088 | 0.19 | 0.05–0.65 |
| | LI vs RH | 1.96 | 0.70 | 0.0068 | 7.13 | 1.75–29.04 |
| | LI vs RI | 0.69 | 0.66 | 0.2972 | 2.00 | 0.54–7.48 |
| | RH vs RI | −1.27 | 0.66 | 0.0565 | 0.28 | 0.08–1.04 |

**Sample quality: ICC** (logit scale): **0.02** (Dog-level variance = 0.06); **Iron stores: ICC** (logit scale): **0.70** (Dog-level variance = 7.54)

Odds ratios (OR) refer to the odds of a higher score for the first site in the contrast versus the second. Values <1 indicate lower odds of a higher score for the first site.

inter-individual differences in marrow composition across the dogs, and highlighting that anatomic location and individual factors jointly shape the quality and histologic interpretation of bone marrow core biopsies in clinical practice.

## Discussion

This randomized clinical trial provides compelling evidence that multi-site bone marrow core sampling significantly improves diagnostic accuracy compared to the current veterinary standard of single-site biopsy. [6,7,10] The study demonstrates clear benefits to sampling multiple anatomical sites, with diagnostic capture probability increasing substantially from one to two sites and continuing to improve with additional sites. These findings fundamentally challenge established veterinary practice and align with emerging recognition of spatial heterogeneity in hematopoietic diseases. [16–18,22,25]

The most striking finding was the significant improvement in diagnostic capture when moving from one to two sampling sites. Under the ANY rule (one pathologist correct), the probability increased from 76.6% to 94.8%, representing an 18.2% improvement that was highly significant (P < 0.0001). Even under the more stringent BOTH_same rule (both pathologists agree on a correct diagnosis), the probability increased from 28.1% to 47.9%, a 19.8% improvement. These substantial gains with the addition of just one additional site underscore the significant risk of diagnostic error when relying on single-site sampling, consistent with recent evidence highlighting sampling limitations in selected human bone marrow pathologies. [23,31–33]

The observed benefit aligns with findings from human oncology, where multi-site tumor sampling has been shown to outperform routine single-site approaches in detecting intratumoral heterogeneity and high-grade disease components. [20,21,23] When heterogeneity is regionally rather than randomly distributed, as is characteristic of hematopoietic diseases, sampling multiple sites provides substantially greater diagnostic information than collecting larger volumes from a single location. [17,20,22,24]

Analysis of secondary variables revealed significant site-dependent differences that further strengthen the case for multi-site evaluation. Core biopsies from the iliac crest consistently yielded samples of higher quality and contained greater iron stores compared to those from the humerus (p = 0.002 for sample quality; p = 0.004 for iron stores). This finding may have critical clinical implications, as suboptimal sample quality can render specimens non-diagnostic, while accurate iron assessment is essential for evaluating iron deficiency anemia, chronic inflammatory conditions, and other

systemic disorders. Site differences between the iliac crest and humerus may reflect anatomical and physiological differences between flat and long bones. The iliac crest contains predominantly cancellous bone with extensive hematopoietic tissue, while the humerus may require greater penetration to reach the medullary cavity. This aligns with findings in human medicine, where site-specific differences in cellular composition are also well-documented. However, the superior site may vary, as seen in humans, where humeral sites sometimes yield higher mesenchymal stem cell concentrations than the iliac crest. [13,28,29,34–37]

Substantial inter-dog variability in iron stores (ICC = 0.70) was observed, reflecting biological diversity and differences related to disease status among dogs. These findings indicate that, beyond anatomical site effects, individual patient factors also influence marrow composition. Nonetheless, the significant site-dependent differences identified in our study demonstrate that sampling location itself contributes independently to diagnostic variability, supporting the value of multi-site evaluation. [29,38,39]

The modest overall inter-pathologist agreement (κ = 0.30, 42% agreement) observed in our study is consistent with reported variability in human bone marrow pathology, where inter-observer agreement ranges from poor to moderate depending on the specific diagnostic criteria and disease entity. [40–42] Site-specific variation in agreement (highest at right humerus, lowest at left humerus) suggests that anatomical location may influence diagnostic confidence, adding another dimension to the argument for multi-site sampling. These findings mirror challenges documented in human hematopathology, where substantial inter-observer variability has been reported for morphologic assessments of dysplasia, blast counts, and specific disease classifications. [31,33] If certain anatomical sites are inherently more difficult to interpret accurately, whether due to technical factors during sampling, [38,39] processing artifacts, or inherent tissue characteristics, multi-site sampling provides opportunities to capture more diagnostically reliable material. [10,19,34,38,39,43–46]

The concept of spatial heterogeneity in bone marrow is increasingly recognized in both human and veterinary medicine. [16–18,22,25,28,47] Recent advances in spatial transcriptomics have revealed complex microenvironmental gradients within bone marrow, with distinct cellular niches and signaling domains. Hematopoietic stem cells show preferential localization to specific anatomical regions, and malignant cells can exhibit spatially restricted distributions. [16,17,22,48] For example, in multiple myeloma, studies have demonstrated that plasma cells are not homogeneously distributed throughout the bone marrow, with osteolytic lesions representing areas of concentrated infiltration that may contain biologically distinct cellular populations. [20,21,49,50] Similarly, clonal hematopoiesis studies have identified intra-patient spatial heterogeneity, with mutant clones detected at one anatomical location but not at contralateral sites. [24,25,51] This spatial organization likely reflects the complex architecture of bone marrow microenvironments, where different anatomical sites may support distinct cellular populations and disease processes. The observed differences in sample quality and iron content between sites in our study may represent manifestations of this underlying spatial organization of hematopoietic tissue.

The findings of this study have immediate practical implications for veterinary hematologic diagnosis. The current standard of single-site sampling carries a significant risk of false-negative results, particularly for patchy or regionally distributed diseases. [20,21,23] The 18.2% improvement in diagnostic accuracy achieved by adding a second site represents a clinically meaningful enhancement that could substantially impact patient outcomes. The site-specific differences in sample quality also inform optimal sampling strategies. Based on site-specific differences, the combination of one iliac and one humeral site appears most advantageous. Iliac crest biopsies provided higher sample quality and iron content, whereas humeral sites demonstrated slightly greater diagnostic correctness. Sampling from both a flat (ilium) and a long bone (humerus), therefore, captures complementary marrow characteristics and likely optimizes diagnostic yield when a two-site approach is used. [13,28,35] From a procedural standpoint, multi-site sampling using powered biopsy systems appears feasible and well-tolerated, as evidenced by the successful completion of four-site sampling in all study dogs. [14,43–46] The incremental time and cost associated with additional sites must be weighed against the substantial diagnostic benefits [50,51] demonstrated in this study.

Several limitations should be acknowledged when interpreting these results. First, the relatively small sample size (16 dogs) limits the statistical power for detecting smaller effect sizes, particularly for subgroup analyses of specific disease entities. The study population was also restricted to dogs requiring bone marrow evaluation for clinical indications, which may not represent the full spectrum of hematopoietic diseases encountered in veterinary practice. Second, the "truth set" methodology used to assess diagnostic accuracy, while innovative in the absence of an external gold standard, relies on the assumption that the most frequent diagnosis across sites and pathologists represents the correct diagnosis. This approach may potentially bias results toward more common conditions or may not account for cases where different sites genuinely harbor different pathologic processes. Furthermore, routine bone marrow histopathology evaluated without immunophenotyping or molecular assays has inherent limitations for resolving lineage specificity (myeloid vs lymphoid) and definitively distinguishing acute leukemia/myelodysplastic syndrome from other round-cell neoplasms (e.g., lymphoma, plasma-cell neoplasia). Conversely, applying diagnosis-specific diagnostic weighting or reclassifying cases post-hoc would introduce information bias and circular reasoning, as these corrections would use the same histologic data that required reclassification, creating a methodological loop that cannot be validated in the absence of an external reference standard. Third, the study was conducted at a single institution with a standardized protocol and an experienced operator, which may limit generalizability to other clinical settings with different expertise levels or equipment. The learning curve associated with multi-site sampling techniques and potential variations in sample processing across institutions could influence the reproducibility of these findings. [32,38,39,43–46] Fourth, while this study has proven that multi-site biopsy is better than a single biopsy from a single site, it was not designed to assess whether multi-site biopsy is better than multiple biopsies from the same anatomical site (an important future study). However, our data and biological rationale still support prioritizing two different sites when a second core is taken, because within-dog analyses showed meaningful across-site discordance in correctness, and because sampling two distinct compartments better addresses spatial heterogeneity than repeating the same site. Finally, the clinical outcomes associated with the improved diagnostic accuracy demonstrated in this study were not factored into the study design and were not assessed. While higher diagnostic yield intuitively suggests better patient care, future studies should evaluate whether multi-site sampling translates to improved therapeutic decision-making and patient outcomes.

This study demonstrated that moving from one to two sites materially increases the probability of capturing a correct diagnosis, and supports choosing two distinct anatomic sites to maximize complementary yield. The substantial improvement in diagnostic capture probability, coupled with site-specific differences in sample quality and iron content, demonstrates that anatomical location significantly influences both the adequacy and interpretability of bone marrow specimens. These findings align with emerging understanding of BM spatial heterogeneity and support consideration of multi-site sampling as a best-practice approach to improve diagnostic accuracy in veterinary bone-marrow evaluation. The modest additional procedural complexity appears justified by the significant diagnostic benefits, potentially leading to more accurate diagnoses and improved patient management in canine hematologic diseases.

## Supporting information

**S1 Table. List of Diagnostic Codes.**
(XLSX)

**S2 Table. Diagnosis.**
(XLSX)

## Acknowledgments

The authors gratefully acknowledge the Small Animal Internal Medicine Technicians at the University of Illinois for helping with the bone marrow procedures

## Author contributions

**Conceptualization:** Sara L Connolly, Arnon Gal.

**Data curation:** Arnon Gal.

**Formal analysis:** Sara L Connolly, Rose Raskin, Nicolas Lopez-Villalobos, Arnon Gal.

**Funding acquisition:** Sara L Connolly, Amy N Schnelle, Arnon Gal.

**Investigation:** Sara L Connolly, Rose Raskin, Arnon Gal.

**Methodology:** Sara L Connolly, Michael F Rosser, Amy N Schnelle, Arnon Gal.

**Project administration:** Kristi M Smiley, Arnon Gal.

**Resources:** Arnon Gal.

**Supervision:** Arnon Gal.

**Writing – original draft:** Kristi M Smiley, Arnon Gal.

**Writing – review & editing:** Kristi M Smiley, Sara L Connolly, Rose Raskin, Michael F Rosser, Amy N Schnelle, Nicolas Lopez-Villalobos, Arnon Gal.

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
