## [Decision Letter · Decision Letter 0]

14 Oct 2025

PONE-D-25-50618

Multi-Site Bone Marrow Core Biopsy Improves Diagnostic Accuracy in Dogs with Hematologic Disease

PLOS ONE

Dear Dr. Gal,

Thank you for submitting your manuscript to PLOS ONE. After careful consideration, we feel that it has merit but does not fully meet PLOS ONE’s publication criteria as it currently stands. Therefore, we invite you to submit a revised version of the manuscript that addresses the points raised during the review process.

Both Major and minor revisions are required and I urge you to consider your edits and responses very carefully.

Please note that Plos! has a 2 editorial round policy which means that all corrections and edits must be made within 2 rounds of revision.

Please submit your revised manuscript by Nov 28 2025 11:59PM. If you will need more time than this to complete your revisions, please reply to this message or contact the journal office at plosone@plos.org. Please include the following items when submitting your revised manuscript:

We look forward to receiving your revised manuscript.

Kind regards,

Zivanai Cuthbert Chapanduka, MBChB (M.D)

Academic Editor

PLOS ONE

Journal Requirements:

“The authors received funding for this work from the University of Illinois Urbana-Champaign Companion Animal Research Grant, Wayne and Josephine Spangler Fund.”

5. In the online submission form, you indicated that “ The data used for analysis is available upon request”.

6. PLOS requires an ORCID iD for the corresponding author in Editorial Manager on papers submitted after December 6th, 2016. Please ensure that you have an ORCID iD and that it is validated in Editorial Manager. To do this, go to ‘Update my Information’ (in the upper left-hand corner of the main menu), and click on the Fetch/Validate link next to the ORCID field. This will take you to the ORCID site and allow you to create a new iD or authenticate a pre-existing iD in Editorial Manager.

7. Your ethics statement should only appear in the Methods section of your manuscript. If your ethics statement is written in any section besides the Methods, please delete it from any other section.

8. Please ensure that you refer to Figures 1-3 in your text as, if accepted, production will need this reference to link the reader to the figure.

Additional Editor Comments:

Thank you for submitting your very interesting work. The reviewers are unanimous that major revision is required.

Kindly read their comments carefully and comply or rebutt as you see fit. Referenced rebuttals are usually more effective and aid in speeding up the review process. Kindly note that Plos1 operates on a 2 review cycle limit system.

Thank you

Kind regards

Reviewers' comments:

Reviewer's Responses to Questions

**Comments to the Author**

1. Is the manuscript technically sound, and do the data support the conclusions?

Reviewer #1: Partly

Reviewer #2: Yes

2. Has the statistical analysis been performed appropriately and rigorously?

Reviewer #1: Yes

Reviewer #2: Yes

3. Have the authors made all data underlying the findings in their manuscript fully available?

Reviewer #1: No

Reviewer #2: Yes

4. Is the manuscript presented in an intelligible fashion and written in standard English?

Reviewer #1: Yes

Reviewer #2: Yes

5. Review Comments to the Author

Reviewer #1: Thank you for the opportunity to review the manuscript entitled "Multi-Site Bone Marrow Core Biopsy Improves Diagnostic Accuracy in Dogs with Hematologic Disease".

This research aimed to evaluate diagnostic yield of single compared to multiple bone marrow biopsy sites in dogs. They found that increasing from one to two biopsy sites produced the largest increase in diagnostic yield and that quality of samples was better from iliac sites than humeral sites. There were site specific differences in myeloid-to-erythroid ratio and megakaryocyte counts and iron stores were higher in the ilium. There was fair agreement on the diagnoses on each biopsy core between the two pathologists that reviewed the marrows. There was generally fair to moderate agreement on the diagnoses for each pathologist across different anatomical sites of biopsy.

There are some concerns about the methodology and definitions of the study, which require attention. Please see the attached document for details.

Reviewer #2: 1.“limitations of single-site sampling have been particularly evident in human hematologic oncology, where multi-region sampling approaches have shown superior diagnostic yields compared to conventional single-site methods.[23]” gives the impression that multi-site sampling is the norm in humans. “However, despite these advances in human medicine, veterinary hematology has largely maintained single-site sampling protocols” Again, mentioned in the Discussion “The observed benefit aligns with findings from human oncology, where multi-site tumor ampling has been shown to outperform routine single-site approaches in detecting intratumoral heterogeneity and high-grade disease components. [20, 21, 23]” Whilst this data is correct, these few studies have not impacted routine practice in humans. This thread of canine medicine lagging behind humans for BMAT sampling is inaccurate and requires correction. Marrow infiltration in humans has also evolved to incorporate imaging (PET-CT). Whilst not a focus in this paper, has this been considered in dogs?

2.The reasoning for the randomization of the site of BM aspiration in the methodology needs to be clearly explained.

3.Where there any failed marrow aspirate and trephine samples? Did the non-diagnostic reads include failed samples?

4.“However, the substantial inter-dog variability observed for iron stores (ICC=0.72) suggests that patient-specific variables, including the underlying disease state of each dog, also significantly influence sample characteristics. This variability emphasizes that sample quality is determined by both anatomical location and patient-specific factors, highlighting why multi-site sampling provides a crucial buffer against sampling inadequacy from any single location.[29, 38, 39]” This paragraph in the Discussion requires re-wording. It is expected that inter-dog and underlying disease will affect findings (including iron stores). So, the link between patient factors and multi-site sampling is unclear. Much of this paragraph is unnecessary.

5.The secondary objectives were to determine optimal number and sample site combinations. Whilst the site sample number is clear addressed in the Discussion, the site combination discussion is more vague. Was the study able to determine an optimal site combination?

6.“These findings align with emerging understanding of bone marrow spatial heterogeneity and suggest that multi-site sampling should be considered as a new standard of care for veterinary bone marrow evaluation” The sample size is too small to justify this conclusion; would suggest re-wording.

6. PLOS authors have the option to publish the peer review history of their article (what does this mean?). If published, this will include your full peer review and any attached files.

**Do you want your identity to be public for this peer review?** For information about this choice, including consent withdrawal, please see our Privacy Policy.

Reviewer #1: No

Reviewer #2: **Yes:** Dr Nadine Rapiti

---

## [Author Response · Author response to Decision Letter 1]

11 Dec 2025

The authors thank the Reviewers and Editor for the time and effort put into providing critique to improve our manuscript.

Reviewer #1: Thank you for the opportunity to review the manuscript entitled "Multi-Site Bone Marrow Core Biopsy Improves Diagnostic Accuracy in Dogs with Hematologic Disease".

AUTHORS’ RESPONSE: We appreciate your thoroughness and rigor in critically evaluating our manuscript, and we have implemented many of your suggestions. Notably, we really appreciated the idea to add the Table that captures the Truth Set, the Truth Rules, and the Sample Quality and Iron. In doing so, we discovered that our initial secondary-aims model inadvertently included nondiagnostic reads; we have now re-run those analyses excluding nondiagnostic cores, to match the primary analysis set, which changed the absolute values but not the direction of the results from our secondary-aims models.

This research aimed to evaluate diagnostic yield of single compared to multiple bone marrow biopsy sites in dogs. They found that increasing from one to two biopsy sites produced the largest increase in diagnostic yield and that quality of samples was better from iliac sites than humeral sites. There were site specific differences in myeloid-to-erythroid ratio and megakaryocyte counts and iron stores were higher in the ilium. There was fair agreement on the diagnoses on each biopsy core between the two pathologists that reviewed the marrows. There was generally fair to moderate agreement on the diagnoses for each pathologist across different anatomical sites of biopsy. There are some concerns about the methodology and definitions of the study, which require attention. Please see the attached document for details.

Major:

The authors acknowledged in the introduction that bone marrow aspirate and biopsy is required for bone marrow evaluation. However, it appears as if only bone marrow biopsy was performed in this study. Is that correct? And why was an aspirate not performed? Certain pathologies (e.g. MDS) can be diagnosed with greater accuracy if combined aspirate and trephine are evaluated.

AUTHORS’ RESPONSE: Thank you for this important comment. You are correct that the present manuscript focuses exclusively on bone marrow core biopsies. Bone marrow aspirates were also collected from each dog during the same procedure, but the cytologic findings and their correlation with biopsy results are presented in a separate manuscript currently under review in another journal. Combining both datasets in a single paper would have substantially exceeded the scope and length limits of this study. We considered the biopsy component independently appropriate for this analysis, as trephine sections represent a distinct and complementary diagnostic modality that allows evaluation of marrow architecture, fibrosis, and infiltrative disease, parameters that cannot be reliably assessed cytologically. The companion manuscript addresses the complementary cytologic data, ensuring that each modality is examined in adequate depth.

Iron grading is typically performed on bone marrow aspirate and graded according to Gale’s scale from 0-6. What iron grading system was used to grade iron on histology – I see it was grade 1-4, but there isn’t a reference to this grading system. Either it should be referenced or each grade explained.

AUTHORS’ RESPONSE: Thank you for this thoughtful comment. In veterinary medicine, the assessment of iron stores in cytologic marrow samples is far less defined than in human medicine. The Gale method has been evaluated in low numbers of animals in two recent studies (2021, doi: 10.1111/vcp.12947; 2023, doi: 10.1111/vcp.13209). Due to the small physical size of many patients, it is relatively common that marrow particles of insufficient number and/or size are obtained for application of the Gale method. Additionally, Prussian Blue staining of marrow samples is not automatically performed on veterinary bone marrow cytology specimens. Consequently, evaluation and scoring of iron stores on cytology remain subjective and ill-defined, without consistent acceptance and application of a given method.

To our knowledge, the Gale system has not been validated for histologic assessment of core biopsy sections in veterinary species, including dogs. Iron distribution and staining characteristics differ substantially between aspirate smears and decalcified trephine sections. Indeed, a J Clin Pathol study (2005; DOI: 10.1136/jcp.2004.017038) comparing iron staining in aspirates and decalcified trephine biopsies from 155 human bone marrow specimens demonstrated that aspirate smears reflect marrow iron stores more reliably than decalcified sections, likely due to iron loss during decalcification. In that study, a 0–4 grading scale was applied for histologic sections. Similarly, we employed a 4-point ordinal rubric as a study-specific adaptation to enhance reproducibility and facilitate direct site-to-site comparisons within dogs. Each grade (1–4) corresponded to increasing iron deposition across 10 random 40× microscopic fields, and the operational definitions for these categories have now been provided in the Methods for clarity:

“…1-4 ordinal iron stores score (1=absent; 2=minimal [rare, small foci of faintly visible granules]; 3=moderate [clearly visible multifocal deposits in macrophages or along trabeculae]; 4=abundant [dense, coalescing granules widely distributed throughout the marrow]; based on 10 random 40× microscopic fields)…”

Were any dogs iron-deficient? Was there agreement between sites and pathologists for an iron deficiency diagnosis? Or could that diagnosis not be made because there was no code for it in the code list?

AUTHORS’ RESPONSE: Thank you for this comment. The present study was not designed to assess specific disease entities such as iron deficiency but rather to evaluate the diagnostic capture probability and site-specific variability of bone marrow core biopsy interpretation in dogs with suspected hematologic disease. As outlined in our stated objectives, the focus was on assessing how the number and anatomical distribution of trephine biopsy sites influence diagnostic accuracy, sample quality, and inter-pathologist agreement.

The diagnosis of iron deficiency typically requires integration of multiple parameters, including Prussian blue–stained bone marrow aspirates (doi: 10.1136/jcp.2004.017038; doi:10.1111/j.1751-553X.2008.01100.x), serum iron indices, and erythrocyte morphology. In contrast, iron evaluation on decalcified trephine core sections is limited by both technical factors (iron loss during decalcification) and biologic variability (spatial heterogeneity of iron deposition across marrow regions). As a result, biopsy-based iron grading alone cannot distinguish true systemic iron deficiency from physiologic variability in local iron storage. Therefore, a specific diagnostic code for iron deficiency was not included in the code list (Table 2a), as this condition was outside the intended scope of our study.

The diagnostic code list lists is a mixture of BM findings (hyperplasia, hypoplasia, BM inflammation, BM fibrosis) and real diagnoses (MDS, leukaemia, BM toxicity, metastaic neoplasia). BM findings are not diagnoses and rather point to a diagnosis that must still be made, e.g. BM hyperplasia may be reactive to infection, haemolysis or nutritional deficiency or may be clonal due to MDS or myeloproliferative neoplasm. Similarly, BM toxicity may present as either BM inflammation or BM hypoplasia depending on the toxin; both of these findings also on the code list. How was BM hyperplasia, BM hypoplasia, BM inflammation and BM toxicity defined? Were acute leukaemia and MDS defined according to WHO-HAEM5? Were each of these codes defined for the pathologists reviewing the marrows? I wonder whether the fact that there are both BM findings and BM diagnoses on the code list can explain the low (fair) agreement between pathologists, however there isn’t enough data provided currently to the reader (see next point). There is also no diagnostic code for BM infection, despite one of the indications for BM examination being suspected BM infection.

AUTHORS’ RESPONSE: Thank you for this thoughtful comment. Our code list was intentionally designed as a morphology-driven classification appropriate for trephine core biopsy interpretation in dogs. In routine veterinary reporting, pathologists often record both pattern-level morphologic impressions (e.g., hyperplasia, hypoplasia, inflammation, fibrosis) and disease-level categories (e.g., leukemia/round-cell neoplasia, metastatic neoplasia). Because this study evaluated how the number and anatomical distribution of histologic core biopsies (with CBC) affect diagnostic capture, sample quality, and inter-pathologist agreement, rather than performing full disease work-ups, we adopted a concise code set that could reflect the spectrum of patterns seen on decalcified sections while capturing clear disease entities, avoiding excessive granularity that would underpower site-level comparisons. The two blinded board-certified pathologists were provided a written codebook (now added as Supplementary Table S2) with operational definitions and coding rules. They were instructed to prioritize disease-level codes when definitive features were present and use pattern-level codes when findings were morphologic but non-definitive. Leukemia/round-cell neoplasia and MDS were separated morphologically using blast burden and architectural/cytologic features on cores plus CBC; however, we did not apply full WHO-HAEM5 diagnostic criteria because immunophenotyping, cytogenetics, and molecular testing were beyond the scope of this sampling-strategy study and are mostly unavailable for use in dogs. We recognize that mixing pattern- and disease-level categories can modestly lower inter-observer agreement in morphology-only assessments, but our primary endpoint, diagnostic capture as a function of site number/combination, is robust to such granularity. A separate “BM infection” code was not included because definitive infection requires ancillary confirmation; suspected infectious cases on morphology were captured under “BM inflammation,” and Table 2a has been revised to state this explicitly. For transparency, Methods now reference Supplementary Table S2 (Morphologic Codebook) summarizing the definitions and rater instructions.

The descriptive data of the population is not included in the publication making it somewhat difficult to understand the findings. I suggest the following table.

Dog Path Diagnosis Iron grade Quality

 LI RI LH RH Truth set LI RI LH RH Average LI RI LH RH

1 1 1 1 1 1 1,5 3 3 4 3 3 Good Good Poor Poor

 2 5 5 5 5 2 3 4 4 Good Good Poor Poor

2 1

 2

3 etc. 1

 2

Proportion of poor/very poor quality 0% 100%

AUTHORS’ RESPONSE: We generated SUPP TABLE S1 per Reviewer request. This has been an invaluable idea.

The authors acknowledged that creation of the truth set had the limitation of being created by the most frequently used diagnostic codes for a dog (across all 4 sites and 2 pathologists). However, it would be more important to be able to pick up a real diagnosis (like a lymphoma), even if only at one site. Could the truth set be weighted so that a true diagnosis is included in the truth set (over a BM finding).

AUTHORS’ RESPONSE: Thank you for this thoughtful point. Our aim in defining the truth set was to avoid circularity in the absence of a gold standard. We therefore used an unsupervised, frequency-based dog-level consensus across all 8 reads (4 sites × 2 pathologists). This behaves like a pre-test probability: it selects the label(s) most supported by the totality of reads without assigning diagnosis-specific priors. Methodologically, this limits incorporation bias, reduces the influence of site-specific outliers (e.g., a single atypical field on one slide), and keeps performance estimates comparable across categories. By contrast, forcing “true diagnoses” (e.g., lymphoma) into the truth set whenever they appear at one site effectively introduces supervision through an informative prior. That approach increases sensitivity to focal disease but also risks false positives and spectrum/verification bias, especially when slide quality varies, and requires post-hoc choices about which codes qualify as “true.”

The authors allude to the understanding that certain pathologies have patchy infiltration in the marrow. However, it is less clear whether the authors understand that this only pertains to certain disease entities. For example, aplastic anaemia, acute leukaemia, myeloproliferative neoplasms and myelodysplastic syndrome should be present in all of the marrow sites as by nature they are stem cell diseases, and for these disease entities a single biopsy is generally acceptable. However infiltrating diseases such as myeloma, lymphoma, granuloma and metastatic solid tumours have patchy infiltration requiring multiple biopsies. Even with that being said, the multiple biopsies are not necessarily required from multiple anatomical sites, as long as >20mm of marrow is collected. https://pubmed.ncbi.nlm.nih.gov/12562655/. It must come across in the introduction that multiple biopsy sites is one of the methods used to improve diagnostic yield (but not the only).

AUTHORS’ RESPONSE: The relevant part in the introduction has been revised as follows to address the Reviewer’s comment: “The limitations of single-site sampling have been highlighted in human hematologic oncology, where a limited number of studies demonstrated that multi-region sampling can improve the detection of focal or spatially heterogeneous disease compared to conventional single-site methods.[23] These findings support the broader understanding that hematopoietic disorders may exhibit regionally distributed lesions rather than uniform involvement, suggesting that multi-site sampling could enhance diagnostic accuracy in selected human conditions.[24, 25] In veterinary hematology, however, single-site sampling protocols remain standard practice,[10] and the potential diagnostic benefit of multi-site approaches for canine bone marrow evaluation has not been systematically investigated.[12, 13, 26]”

On this point, the diagnostic codes for leukaemia and round cell neoplasia were combined into one, despite having different marrow infiltration patterns. Would it be possible to separate those entities and report them separately? It would be interesting to see whether acute leukaemia and MDS were indeed found across all biopsy sites, as expected.

AUTHORS’ RESPONSE: We appreciate the interest in site-distribution patterns. However, separating “leukemia” from “round-cell neoplasia” in our dataset is not methodologically defensible. This study deliberately limited interpretation to H&E morphology without immunophenotyping, flow cytometry, PARR testing, or cytogenetics. Under these conditions, canine marrow cannot be reliably subclassified into myeloid vs lymphoid lineages (or acute leukemia vs lymphoma/plasma-cell neoplasia) at the level the Reviewer requests. Implementing a post-hoc split would therefore create false precision, increase misclassification risk, and introduce incorporation/circularity bias by effectively re-labeling cases using assumptions we cannot verify with the routine histopathology.

Our codebook pre-specified a single category (Code 3) for “leukemia/round-cell tumor” precisely to preserve reproducibility across readers using routine stains. The aim of this work was to evaluate cross-site and cross-reader consistency under routine histomorphologic conditions, not to adjudicate disease-specific biology that requires orthogonal methods. Splitting Code 3 after seeing the data would be post-hoc and statistically underpowered, and would undermine comparability across sites and dogs.

For transparency, Supplementary Table S1 (the “Dog × Path × Site” matrix) already shows that Code 3 appears in both multi-site and site-limited patterns across dogs, for example, some dogs have Code 3 at multiple sites while another site in the same dog is non-3, illustrating that mixed topography occurs in practice on H&E and cannot be mapped cleanly to “diffuse vs patchy” entities without im

---

## [Decision Letter · Decision Letter 1]

19 Jan 2026

PONE-D-25-50618R1
Multi-Site Bone Marrow Core Biopsy Improves Diagnostic Accuracy in Dogs with Hematologic Disease
PLOS One

Dear Dr. Gal,

Thank you for submitting your manuscript to PLOS ONE. After careful consideration, we feel that it has merit but does not fully meet PLOS ONE’s publication criteria as it currently stands. Therefore, we invite you to submit a revised version of the manuscript that addresses the points raised during the review process.
 
I am confident that if you address the concerns of one of the 2 peer reviewers, your manuscript will be ready for acceptance. The other peer reviewer recommends acceptance.

We look forward to receiving your revised manuscript.

Kind regards,

Zivanai Cuthbert Chapanduka, MBChB (M.D)

Academic Editor

PLOS One

Journal Requirements:

Reviewers' comments:

Reviewer's Responses to Questions

**Comments to the Author**

1. If the authors have adequately addressed your comments raised in a previous round of review and you feel that this manuscript is now acceptable for publication, you may indicate that here to bypass the “Comments to the Author” section, enter your conflict of interest statement in the “Confidential to Editor” section, and submit your "Accept" recommendation.

Reviewer #1: All comments have been addressed

Reviewer #2: All comments have been addressed

2. Is the manuscript technically sound, and do the data support the conclusions?

Reviewer #1: Yes

Reviewer #2: Yes

3. Has the statistical analysis been performed appropriately and rigorously? 

Reviewer #1: Yes

Reviewer #2: Yes

4. Have the authors made all data underlying the findings in their manuscript fully available?

Reviewer #1: Yes

Reviewer #2: Yes

5. Is the manuscript presented in an intelligible fashion and written in standard English?

Reviewer #1: Yes

Reviewer #2: Yes

6. Review Comments to the Author

Reviewer #1: Thank you for the excellent explanations in the response to the reviewers. The following information was explained in the response to the reviewer, but is not in the methods and would interest the reader or enhance clarity:

- The pathologists were experienced veterinary clinical pathologists

- Immunohistochemistry was not performed

Given the new information provided with the supplementary tables, I have a couple of queries:

Supplementary Table S1:

- In Table S1, there are the columns ANY sites, BOTH_any sites and BOTH_same sites. Do these correlate to the rules ANY, BOTH_any and BOTH_same? If not, then the columns should be labelled differently as they are currently labelled too similar to the rules. I assumed they were referring to the rules for the next comments.

- From the table, it is evident that BOTH_any and BOTH_same was only different for one dog (dog 12), where the truth set had 2 codes. For all other dogs, the truth set had one code and then BOTH_any was the same as BOTH_same. This brings into question the definitions of BOTH_any and BOTH_same.

- Definition of BOTH_any in Table S1 appears to indicate both pathologists were correct at the same site (otherwise dog 13 would have been included as both pathologist had the correct codes, but since the codes were correct at different sites, it was not included). This does not correspond to the BOTH_any definition supplied in the methods (both correct, not necessarily the same code), or in Table 3 (both in truth set). The same definition should be used consistently across different sections/tables and be easy to follow in the dataset.

- Similar applies to BOTH_same (correct with the same code at the same sites, as it appears in Table S1 vs ‘both agree on truth set’ vs ‘both agree and correct’ vs ‘both pathologists agree on a correct diagnosis’). If the site didn’t matter as implied by the definition ‘both agree on the truth set’, then dog 13 should have been included.

Supplementary Table S2:

- For MDS: Can the mentioned dysplastic features be seen on trephine biopsy? Many of those features would typically be evident on bone marrow aspirate and not histology.

Figure 2 and Figure 3:

- The legend is currently unhelpful. The line is presumably the 95% CI and the circle the mean. It should be labelled accordingly.

- As the Figures 2 and 3 contain the same information as Table 3 and Table 5, I suggest omitting the tables.

Table 6 adds limited information. I suggest omitting it and including the 95% CI in the text.

Reviewer #2: (No Response)

7. PLOS authors have the option to publish the peer review history of their article (what does this mean?). If published, this will include your full peer review and any attached files.

Reviewer #1: No

Reviewer #2: **Yes:** Nadine Rapiti

---

## [Author Response · Author response to Decision Letter 2]

28 Jan 2026

The authors thank the Reviewer and Editor for the time and effort put into providing critique to improve our manuscript.

Reviewer #1: Thank you for the excellent explanations in the response to the reviewers. The following information was explained in the response to the reviewer, but is not in the methods and would interest the reader or enhance clarity:

- The pathologists were experienced veterinary clinical pathologists

- Immunohistochemistry was not performed

AUTHORS’ RESPONSE: The information the Reviewer mentioned above is now included in the text.

Given the new information provided with the supplementary tables, I have a couple of queries:

Supplementary Table S1:

- In Table S1, there are the columns ANY sites, BOTH_any sites and BOTH_same sites. Do these correlate to the rules ANY, BOTH_any and BOTH_same? If not, then the columns should be labelled differently as they are currently labelled too similar to the rules. I assumed they were referring to the rules for the next comments.

AUTHORS’ RESPONSE: We changed the column heading to match the rules (i.e., Any, Both_any, and Both_same)

- From the table, it is evident that BOTH_any and BOTH_same was only different for one dog (dog 12), where the truth set had 2 codes. For all other dogs, the truth set had one code and then BOTH_any was the same as BOTH_same. This brings into question the definitions of BOTH_any and BOTH_same.

- Definition of BOTH_any in Table S1 appears to indicate both pathologists were correct at the same site (otherwise dog 13 would have been included as both pathologist had the correct codes, but since the codes were correct at different sites, it was not included). This does not correspond to the BOTH_any definition supplied in the methods (both correct, not necessarily the same code), or in Table 3 (both in truth set). The same definition should be used consistently across different sections/tables and be easy to follow in the dataset.

- Similar applies to BOTH_same (correct with the same code at the same sites, as it appears in Table S1 vs ‘both agree on truth set’ vs ‘both agree and correct’ vs ‘both pathologists agree on a correct diagnosis’). If the site didn’t matter as implied by the definition ‘both agree on the truth set’, then dog 13 should have been included.

AUTHORS’ RESPONSE:

We appreciate the reviewer’s careful reading. The key clarification is that the rules ANY, BOTH_any, and BOTH_same are evaluated within a given site (LI, RI, LH, RH). In other words, a site is counted under a rule only based on the two pathologists’ interpretations at that same site; agreement (or correctness) occurring at different sites does not qualify.

First, a scenario where the “truth set” has ties:

Site:   LI  LH  RI  RH

PATH1:  A  B  A  C

PATH2:  B  C  B  A

TRUTH SET: {A, B} because there are 3 × A and 3 × B

• RULE ANY: LI, LH, RI, RH (each site has at least one call in {A,B})

• RULE BOTH_any: LI, RI (at these sites, both pathologists’ diagnoses fall within the truth set {A,B}, even though they selected different codes).

• RULE BOTH_same: none (there is no site where both pathologists selected the same truth-set code).

This example illustrates why BOTH_any and BOTH_same can differ when the truth set has ties: within a site, the two pathologists can each be “in the truth set” while choosing different tied codes.

Second, a scenario where the “truth set” has no ties:

Site:   LI  LH  RI  RH

PATH1:  A  B  A  C

PATH2:  B  B  B  A

TRUTH SET: {B} because there are 4 × B

• RULE ANY: LI, LH, RI (sites with at least one B)

• RULE BOTH_any: LH (only LH has BB)

• RULE BOTH_same: LH (only LH has BB)

This example shows that when the truth set is a single code, BOTH_any and BOTH_same are necessarily identical, because “both pathologists within the truth set” implies both selected the same single truth-set code.

Therefore, BOTH_any and BOTH_same differ only when (1) the truth set contains ties, and (2) within the same site one pathologist selects one truth-set code while the other selects the other truth-set code.

Regarding Dog 13, the truth set contains a single code (no ties), and there is no site at which both pathologists selected the truth-set code. Although each pathologist selected the truth-set code at some site, they did not do so at the same site, so BOTH_any and BOTH_same are correctly empty for Dog 13 under the within-site definitions.

To ensure consistent terminology across the manuscript, we revised the Methods and the Table 3 footnote to explicitly reflect the within-site nature of these rules:

• ANY: within a site, at least one pathologist is within the truth set.

• BOTH_any: within a site, both pathologists are within the truth set (when the truth set has ties, they may not necessarily choose the same truth-set code).

• BOTH_same: within a site, both pathologists are within the truth set and both choose the same truth-set code (even when there are ties).

According to these definitions, Supplementary Table S1 is correct. We also reordered the BM sampling sites in Supplementary Table S1 to match the order used when listing sites under the “Truth set” (LH, LI, RH, RI) to improve readability.

Supplementary Table S2:

- For MDS: Can the mentioned dysplastic features be seen on trephine biopsy? Many of those features would typically be evident on bone marrow aspirate and not histology.

AUTHORS’ RESPONSE: Thank you for this comment. We agree that several dysplastic features commonly used to support a diagnosis of myelodysplastic syndrome (MDS), particularly subtle erythroid and granulocytic abnormalities, are generally more readily appreciated on bone marrow aspirate cytology than on trephine histology. Supplementary Table S2 lists diagnostic criteria for MDS rather than features that are uniquely or optimally assessed on trephine sections alone.

In the context of trephine evaluation, marrow cellularity and blast proportion (including blast aggregation/clustering) can be assessed on histologic sections, and megakaryocytic abnormalities are often appreciable. While recognition of erythroid and granulocytic dysplasia on histology is more challenging, these features can be identified in trephine sections by experienced hematopathologists, albeit with less sensitivity than on aspirate smears. Accordingly, such features were interpreted cautiously on histology and considered supportive rather than definitive in isolation.

Thus, Supplementary Table S2 reflects standard diagnostic criteria for MDS, whereas the relative ease of identifying individual features depends on the specimen type and was accounted for during interpretation.

Figure 2 and Figure 3:

- The legend is currently unhelpful. The line is presumably the 95% CI and the circle the mean. It should be labelled accordingly.

AUTHORS’ RESPONSE: legends have been revised as follows:

Figure 2. Model-based site effects on per-reading correctness. GLIMMIX least-squares means with 95% CIs for LH, LI, RH, RI on the probability scale. No site effect (p=0.599) and no pathologist effect (p=0.781). Point: LS-mean predicted probability; Horizontal line: 95% CI.

Figure 3. Inter-pathologist agreement by site pair. Simple κ with 95% CIs for each site pair pooled across pathologists, with N overlaid. Highlights that LI–RH shows the highest agreement. Open circle: κ estimate; Horizontal line: 95% CI.

- As the Figures 2 and 3 contain the same information as Table 3 and Table 5, I suggest omitting the tables.

AUTHORS’ RESPONSE:

We thank the Reviewer for this suggestion. While we agree that Tables 3 and 5 and Figures 2 and 3 summarize overlapping results, we respectfully believe that the tables and figures provide complementary, rather than redundant, presentations of the data, and that retaining both enhances clarity for readers with different preferences for data interpretation.

Specifically, the tables provide precise numerical estimates and confidence intervals that allow detailed comparison across conditions, whereas the figures convey the same results in a visual format that facilitates rapid assessment of effect size, uncertainty, and overlap across sites and site pairs. In our experience, these two formats support distinct cognitive approaches to understanding the results, quantitative scrutiny versus pattern recognition, and are commonly used together for this reason.

Because the study addresses spatial heterogeneity and agreement across biopsy sites, we believe the graphical displays are particularly valuable for visually communicating site-level trends and uncertainty that are less immediately apparent from tabular data alone, while the tables remain essential for exact reporting.

For these reasons, we respectfully request that the Reviewer reconsider the suggestion to omit either the tables or the figures. We would also welcome the Editor’s guidance on whether retaining both formats is appropriate for the journal and readership.

Table 6 adds limited information. I suggest omitting it and including the 95% CI in the text.

Authors' Response: Thank you for this suggestion. We have omitted Table 6 and incorporated the key results, including the 95% confidence intervals, into the Results section. The remaining tables have been renumbered consecutively.

---

## [Decision Letter · Decision Letter 2]

3 Feb 2026

Multi-Site Bone Marrow Core Biopsy Improves Diagnostic Accuracy in Dogs with Hematologic Disease

PONE-D-25-50618R2

Dear Dr. Arnon Gal,

We’re pleased to inform you that your manuscript has been judged scientifically suitable for publication and will be formally accepted for publication once it meets all outstanding technical requirements.

Kind regards,

Zivanai Cuthbert Chapanduka, MBChB (M.D)

Academic Editor

PLOS One

Additional Editor Comments (optional):

Reviewers' comments:

Reviewer's Responses to Questions

**Comments to the Author**

1. If the authors have adequately addressed your comments raised in a previous round of review and you feel that this manuscript is now acceptable for publication, you may indicate that here to bypass the “Comments to the Author” section, enter your conflict of interest statement in the “Confidential to Editor” section, and submit your "Accept" recommendation.

Reviewer #1: All comments have been addressed

2. Is the manuscript technically sound, and do the data support the conclusions?

Reviewer #1: Yes

3. Has the statistical analysis been performed appropriately and rigorously? 

Reviewer #1: Yes

4. Have the authors made all data underlying the findings in their manuscript fully available?

Reviewer #1: Yes

5. Is the manuscript presented in an intelligible fashion and written in standard English?

Reviewer #1: Yes

6. Review Comments to the Author

Reviewer #1: All concerns have been addressed and resolved appropriately.

There are no futher comments.

7. PLOS authors have the option to publish the peer review history of their article (what does this mean?). If published, this will include your full peer review and any attached files.

Reviewer #1: No

---

## [Editor Report · Acceptance letter]

PONE-D-25-50618R2

PLOS One

Dear Dr. Gal,

I'm pleased to inform you that your manuscript has been deemed suitable for publication in PLOS One. Congratulations! Your manuscript is now being handed over to our production team.

Kind regards,

on behalf of

Professor Zivanai Cuthbert Chapanduka

Academic Editor

PLOS One